# Accurate measurement of simulated slow and altered walking activity: Apple Watch best in class wearable devices

Grant Rowe[1], David Weight[2,3], Alethea Rea [2], Jenny Conlon[4], Fiona M. Wood[3,5,6], Dale W. Edgar [3,5,6,7]*

1 Discipline of Psychology, College of Science, Health, Engineering and Education, Murdoch University, Perth, Western Australia, Australia, 2 School of Mathematics, Statistics, Chemistry and Physics, Murdoch University, Murdoch, Western Australia, Australia, 3 Fiona Wood Foundation, Fiona Stanley Hospital, Murdoch, Western Australia, Australia, 4 Faculty of Medicine, Nursing, Midwifery and Health Sciences, School of Health Sciences, The University of Notre Dame Australia, Fremantle, Western Australia, Australia, 5 State Adult Burn Unit, Level 4, Fiona Stanley Hospital, Murdoch, Western Australia, Australia, 6 Burn Injury Research Unit, University of Western Australia, Crawley, Western Australia, Australia, 7 Institute for Health Research, The University of Notre Dame Australia, Fremantle, Western Australia, Australia

* dale.edgar@health.wa.gov.au, dale.edgar@nd.edu.au

## Abstract

### Background

Wearable activity devices, widely used to monitor physical activity in non-injured populations, have shown potential in encouraging early ambulation and enhanced recovery in hospitalised patients. This study evaluated the accuracy of wearable devices in tracking step counts under simulated hospital conditions, seeking the optimal body location placement for individuals with altered gait.

### Methods

This method comparison study involved healthy adults walking on a treadmill while performing slow and shuffling walking patterns. Twelve wearable devices were placed on the arm, waist, and leg, and their recorded step counts were compared to manual counts from filmed sessions, following Consumer Technology Association guidelines.

### Results

The Apple Watch, particularly when worn on the waist, demonstrated the highest reliability and adaptability across walking patterns. Leg placement, which accounted for 10 of the top 20 device-position combinations, suggested that larger movement amplitudes improve step count accuracy, particularly during slow or altered gaits.

### Conclusion

This study confirmed the Apple Watch to be the most accurate wearable step count device. The study provides new understanding as to the precision of commercially

**Data availability statement:** Data cannot be shared publicly because of Ethics approval provisions. Data are available from the SMHS Institutional Data Access / Ethics Committee (contact via SMHS.rgo@health.wa.gov.au) for researchers who meet the criteria for access to confidential data.

**Funding:** The author(s) received no specific funding for this work.

**Competing interests:** The authors have declared that no competing interests exist.

available devices and their placement, when aiming to improve and, or conduct research about, patient physical activity outcomes.

## Introduction

Wearable activity devices, commonly used by non-injured populations to monitor motion [1], have been shown to increase activity levels over time [2,3]. Similar benefits may apply in hospitalised patients, where reduced activity with bed rest can itself lead to muscle weakening, joint stiffness, cardiovascular decline [4–6], and heightened risks of complications such as deep vein thrombosis and hospital-acquired infections [7]. These issues, when exacerbated by limited ambulation due to injury or disease, contribute to deterioration in physical and mental health [8]. In contrast, inpatient early ambulation is associated with improved outcomes, including preserved muscle strength [9], better circulation [10], enhanced respiratory function [11], and psychological well-being [12], resulting in quicker recovery, shorter hospital stays, and reduced readmission rates [13–16].

Accurately tracking physical activity during hospitalisation is essential for optimising care. Traditional and time-expensive methods, like manual or observational tracking, have been largely replaced by sensor-based devices such as accelerometers and pedometers, which provide continuous, objective movement monitoring [17]. Wearable activity devices are valued for their lightweight design and ability to track step counts, offering real-time feedback to clinicians and motivational cues to patients [18,19]. However, wearable devices often rely on algorithms and sensors designed for normal walking patterns, and speeds, leading to inaccuracies in patients with altered gaits and, or limited arm movement where devices are primarily applied [20,21].

Despite their potential, the lack of understanding as to the accuracy of wearable activity devices in hospital settings remains a challenge for clinicians and researchers, especially for patients with slow or shuffling gait patterns [22]. This study aimed to evaluate the accuracy and precision of commercially available devices in recording step counts compared to manual counting and to identify the optimal body placement for wearables (arm, waist, or leg). It was hypothesised that in those with altered (slow and shuffle) gaits, activity devices would underestimate step counts (representing overall walk activity) and that devices positioned on the arm would yield the most valid and reliable measurements.

## Methods

This method comparison study assessed the accuracy of wearable activity devices in registering (recording) step count, during both slow and shuffling gait patterns, with devices placed at multiple body positions (arm, waist, and leg). The study design purposefully deployed devices 'not as intended' to test the limits of their capabilities.

### Participants

Twenty-seven non-injured adults (females: n = 14) were recruited from the local hospital staff and community between 18/08/2020 and 31/03/2021. Eligibility criteria included the absence of self-reported health issues, particularly

musculoskeletal conditions, that could impair safe walking during multiple repeats of five minutes on a motorised treadmill. All recruited participants provided written informed consent (hard copy), witnessed by the coordinating researcher (DW). The study protocol adhered to the Declaration of Helsinki and received approval from the University of Notre Dame Human Research Ethics Committee (019176F) and the South Metropolitan Health Service (RGS3971).

## Protocol

**Simulated gait conditions.** All participants attended two 60-minute testing sessions, each comprising six 'blocks' of five-minutes of altered walking on the same motorised treadmill (Landice L7 Cardio, USA) to accommodate all combinations of wearables tested. Each block consisted of a single gait pattern, with speeds selected via the treadmill digital speedometer:

- Slow walk: Walking at 3.5 km/h with normal foot translation with arm swing reciprocal and symmetrical in association with leg speed.

- Shuffle walk: Walking at 2.0 km/h with a bilateral, simulated reduced hip and knee flexion trajectory in swing phase and arm swing as above.

## Data collection

**Wearables.** Participants were not specifically informed of the device brands as and when they were applied at each site, though they were not specifically blinded to this information. During each five-minute walking block, two devices were worn on the wrists (one per side), two on the ankles, and two on the waist. These positions were categorised as arm, leg, and waist, respectively, with a strap securing the waist devices. Across 12 walking blocks (six slow walk, six shuffle walk), devices were systematically rotated so that each occupied all three positions under both gait conditions. Device–position assignments were randomised and counterbalanced across participants using built-in Microsoft Excel functions, ensuring each device was tested once per gait style at each position while minimising potential order effects. Participants were asked to remain as motionless as possible between blocks so that additional steps were not registered during changeover of devices between body sites.

During each session, the participants were fitted with 12 activity devices (for additional details, see S1 File):

- ActiGraph wGT3X-BT*,

- Apple Watch Series 5

- Fitbit Charge 4,

- Fitbit Inspire 1,

- Fitbit Versa 2,

- Garmin Forerunner 45S,

- Garmin Vivofit 4,

- GENEActiv*,

- Samsung Active 2,

- Samsung Fit 1,

- Withings Move,

- Withings Steel HR

The 10 activity devices capable of real-time step displays had their counts recorded at the start and end of each five-minute block to represent the number of steps registered during the walk period, referred to as the estimated step count for each device. In addition, for the two devices (*ActiGraph wGT3X-BT and GENEActiv) that do not display real time counts, the start and end times of each block were documented to synchronise the steps registered from raw data collected on those devices.

**Step count.** To measure the accuracy of step numbers registered on each wearable device, a video recording was taken of each participant's legs (with consent), with one step defined as by either foot striking the ground at the apex of movement cycle at the top of the treadmill. The video was reviewed after collection sessions, and steps counted visually by a single investigator (DW). The visual count was conducted twice and if the counts did not match exactly, repeat counts were conducted until consecutive, exact-match counts were observed and the result was paired to the activity device data (as suggested in the Consumer Technology Association [CTA] guidelines) [23]. No identifying information was relayed to devices or collected during filming, apart from participant study number.

### Data processing – Devices without real time displays*

Step counts were extracted from ActiGraph accelerometers, recorded in 1-minute epochs using ActiLife software (v6.13.4, ActiGraph LLC, USA) and processed in R (v3.5.1) [24]. The chron (v2.3-62) [25] package was used to standardise time formatting and filter the dataset. A 6-minute window was selected from a predefined start time to ensure full capture of the walking block, accounting for potential misalignment in the 1-minute sampling. Step counts were then summed within this window to obtain the total step count. In contrast, GENEActiv accelerometer data, recorded at 100 Hz, was processed using R (v3.5.1) and GENEAclassify (v1.4.18) [26]. The GENEAclassify package was used to analyse.bin files, applying the getGENEAsegments function with the "none" method to prevent predefined segmentation. This approach allowed for a continuous extraction of step counts over the entire recording period, enabling an uninterrupted assessment of walking activity.

### Data analysis

Data analysis was performed using R (v4.4.1). Activity device step count performance was evaluated using percentage difference to assess the magnitude and direction of deviation from the true standard (video based step count), margin of error (MOE) to determine precision (defined as half the width of the 95% confidence interval (CI) for the percentage difference between estimated and true step counts) and mean absolute percentage error (MAPE) to summarise the overall error magnitude. According to the CTA guidelines, MAPE scores below 10% indicate acceptable accuracy [23]. To standardise performance comparisons across metrics with different scales, raw scores were normalised to a 0–1 range, with higher scores indicating better performance. For percentage difference, absolute differences were scaled relative to the maximum observed difference and inverted, so smaller differences scored closer to 1. For MOE and MAPE, scores were adjusted by subtracting the minimum, dividing by the range (maximum – minimum), and inverting to ensure lower errors scored higher. This normalisation allowed consistent comparisons across metrics and enabled a composite evaluation of device performance. Confidence intervals (CIs) were analysed to identify systematic bias, with CIs crossing zero interpreted as indicating no significant bias. These analyses were also repeated to evaluate the overall performance of each device, irrespective of position. To further investigate differences between devices while accounting for repeated measures within participants, a linear mixed-effects model (LMM) was deployed. The outcome variable was step count difference (estimated minus true steps), with fixed effects for Device, Position, and Activity, and a random intercept for Participant to account for within-subject dependencies within the data. Estimated marginal means were used to derive pairwise comparisons between devices, and this procedure was bootstrapped (1,000 resamples) to generate robust 95% confidence intervals for pairwise differences. This comparison analysis was restricted to the four devices with the lowest MAPE values—Apple Watch, Garmin Vivofit, Fitbit Charge, and Samsung Active—which represented the top performers

in earlier error-based evaluations. Data preprocessing, analysis, and visualisation were performed using the following R packages: dplyr [27], tidyr [28], psych [29], ggplot2 [30], RColorBrewer [31], scales [32], reshape2 [33], lmer [34], emmeans [35], and boot [36].

## Results

### Activity device performance across various body positions

Fig 1 illustrates the estimated steps registered by each activity device for each body position during walking (slow and shuffling) on a motorised treadmill for 5 minutes. Fig 2 highlights the percentage difference and 95% CIs between the estimated steps from each device-position combination compared to the true standard (manual counting). Among the 36 device-position combinations, the three with the smallest percentage differences were the Apple Watch positioned on the waist (1.9%), the Samsung Fit positioned on the leg (−2.3%), and the Fitbit Charge positioned on the leg (4.7%), indicating minimal and CTA-acceptable deviation relative to the true standard. In contrast, the GENEActiv positioned on the arm (−91.1%), the GENEActiv positioned on the waist (−67.2%), and the Withings Move positioned on the arm (−56.5%) exhibited the largest percentage differences, reflecting significant deviations from the true standard.

The devices with the smallest margins of error (MOE), as indicated by the narrowest CIs between repeats, were the Apple Watch on the waist (3.4%), the Fitbit Charge on the leg (3.8%), and the Fitbit Versa on the leg (4.2%). These results suggested they would be more reliable if used for repeated measurements. In contrast, the largest MOE values were observed for the GENEActiv on the leg (23.7%), the Garmin Forerunner on the leg (13.8%), and the Withings Steel on the

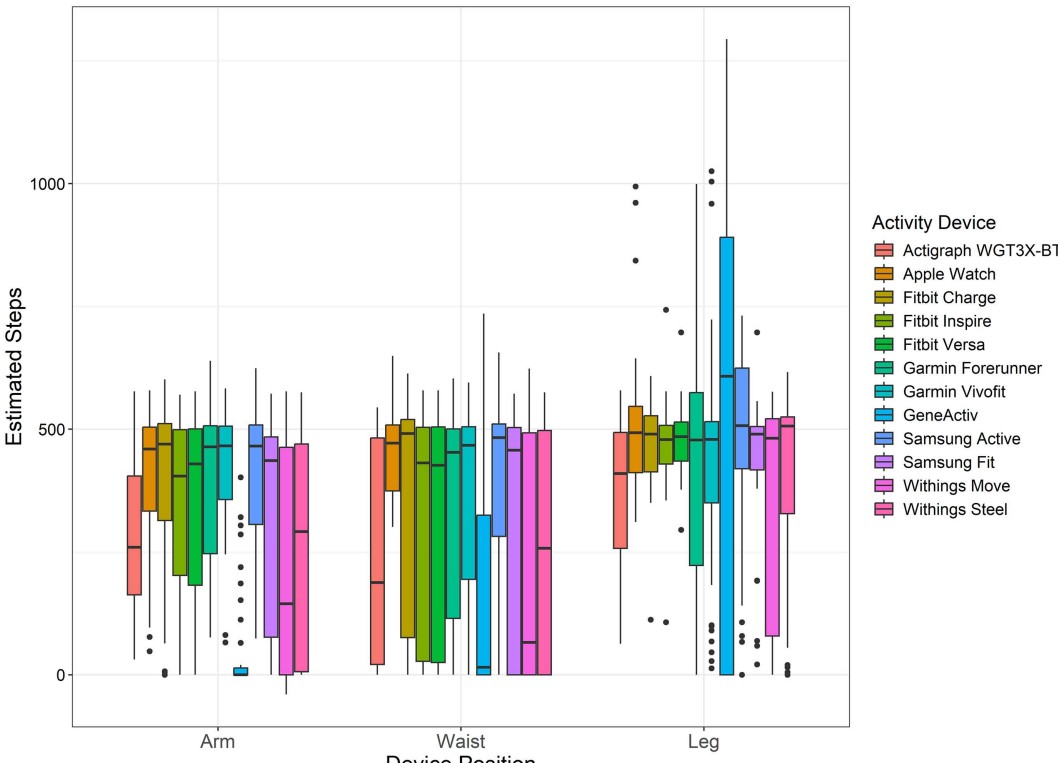

**Fig 1. Estimated steps recorded for each activity device by body position.** Each plot represents the median with interquartile range (25th to 75th percentiles). Whiskers extend up to 1.5 times the interquartile range beyond the hinges. Outliers displayed as filled circles.

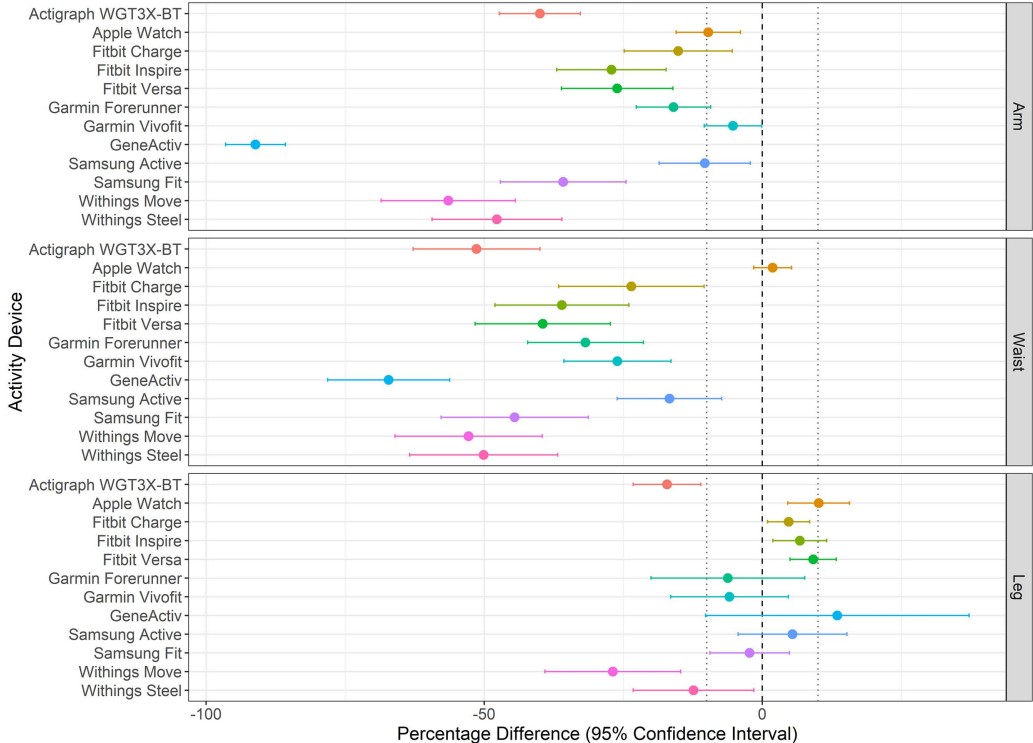

**Fig 2. Percentage difference between device estimated and true step counts by body position.** Vertical dashed lines indicate 10% acceptable threshold (Consumer Technology Association defined) from true count.

waist (13.3%), indicating greater variability and reduced confidence in their precision and consistency if applied across repeated, altered gait assessments (Fig 2).

Fig 3 illustrates the mean absolute percentage error (MAPE) of each activity device at the different body positions relative to the true standard. The Apple Watch positioned on the waist (5.8%), Fitbit Charge positioned on the leg (7.2%), and Fitbit Inspire positioned on the leg (9.7%) exhibited the lowest MAPEs, indicating the highest predictive accuracy. Two additional device-position combinations met the CTA-defined <10% MAPE criterion for acceptable accuracy: the Garmin Vivofit on the arm (9.8%) and the Fitbit Versa on the leg (9.9%). In contrast, the GENEActiv device demonstrated the highest MAPE values across all testing positions (arm: 91.1%; leg: 79.5%; waist: 70.3%), reflecting significant inconsistencies in predictions.

Six activity devices, positioned at various body positions, had percentage difference CIs crossing zero, indicating no systematic bias in estimating the true step counts (Fig 2). These device-position combinations were: (1) Apple Watch on the waist, (2) Garmin Forerunner on the leg, (3) Garmin Vivofit on the leg, (4) GENEActiv on the leg, (5) Samsung Active on the leg, and (6) Samsung Fit on the leg. As hypothesised, most (26) device-position combinations systematically underestimated the true steps, while four devices positioned on the leg systematically overestimated the true steps (Fig 2).

Fig 4 displays a heatmap showcasing the normalised performance scores, including percentage difference, MOE, and MAPE, for each device-position combination. The performance of step count activity devices varied considerably across different body positions, with notable disparities between the best and worst combinations. The Apple Watch positioned at the waist emerged as the top performer, achieving the highest average normalised score of 1.00 across all metrics, making it the most accurate and precise device-position combination. The Fitbit Charge on the leg ranked second overall with an average normalised score of 0.98, followed by the Fitbit Versa on the leg, scoring 0.94. These results highlight the strong tracking capabilities of these devices

**Fig 3. Mean absolute percentage error (MAPE) between device estimated and true step counts by body position.** Vertical dashed line marks 10% acceptance threshold.

in their respective positions. In contrast, the GENEActiv device consistently underperformed across all positions. The GENEActiv on the arm had the lowest overall score (0.30), followed by the leg (0.34) and waist (0.38). Interestingly, some device-position combinations exhibited 'trade-offs' across metrics. For example, the Garmin Forerunner on the leg performed well in terms of percentage difference, reflecting low deviation from the reference standard (high precision), but had poorer MOE and MAPE scores, indicating reduced consistency and predictive reliability. Conversely, the ActiGraph wGT3X-BT on the arm demonstrated relatively high precision but suffered with percentage difference and MAPE indicators, likely due to a systematic underestimation bias.

## Activity device performance irrespective of body position

Fig 5 highlights the percentage difference and 95% CIs between the estimated steps from each device, irrespective of body position, and the true standard. Among the 12 devices, the Apple Watch (1.0%), Samsung Active (−7.2%), and Fitbit Charge (−11.3%) had the smallest percentage differences, indicating minimal deviation from the true standard. In contrast, the GENEActiv (−48.3%), Withings Move (−45.4%), and Withings Steel (−36.7%) displayed the largest percentage differences, reflecting substantial deviations from the true standard.

The devices with the smallest margins of error (MOE), as indicated by the narrowest CIs, were the Apple Watch (3.1%), Garmin Vivofit (5.2%), and ActiGraph wGT3X-BT (5.3%). In contrast, the GENEActiv (11.1%), Withings Move (7.4%), and Withings Steel (7.3%) demonstrated the largest MOE values, indicating lower precision (Fig 5).

Fig 6 illustrates the MAPE of each activity device, irrespective of body position, relative to the true standard. The Apple Watch (9.3%), Garmin Vivofit (19.2%), and Fitbit Charge (22.0%) had the lowest MAPEs, demonstrating the highest predictive accuracy no matter where the activity device was positioned. In contrast, the GENEActiv (80.3%), Withings Move

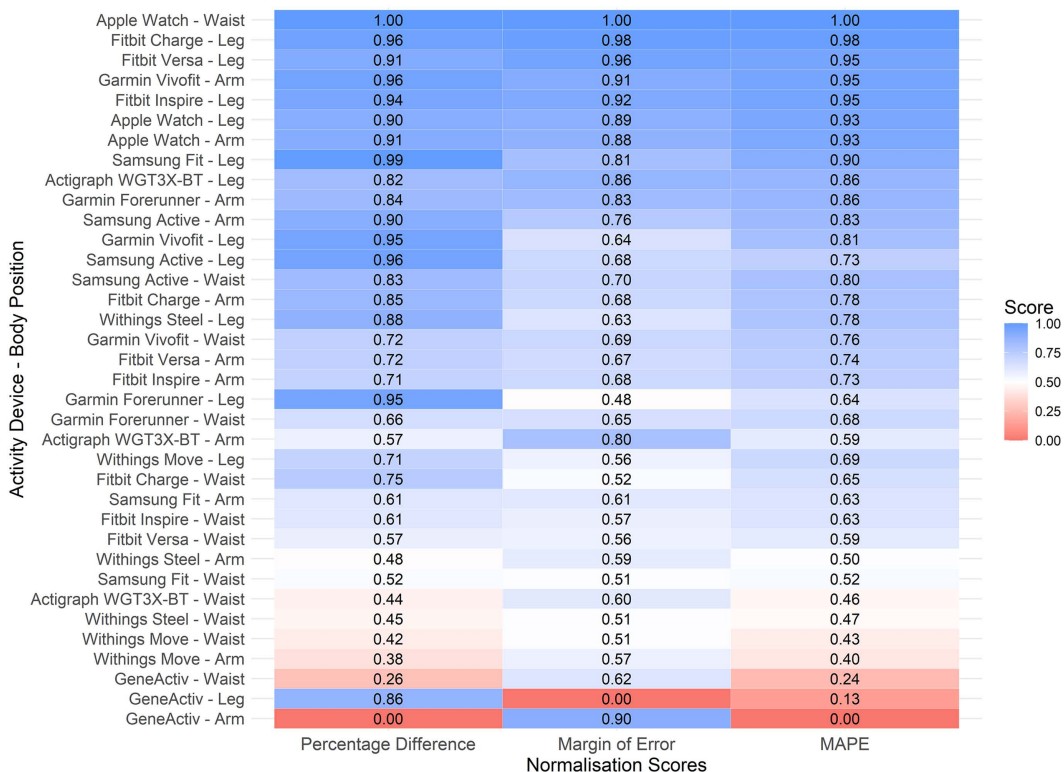

**Fig 4. Heatmap of device performance metrics for activity devices by body position.** Normalised scores (0-1) for: percentage difference (left), margin of error [MOE](middle), and mean absolute percentage error [MAPE](right). Colour shading: lower (red); moderate (white); and, higher (blue) normalised performance scores. Device rank reflects overall performance across device-position combinations.

(47.7%), and Withings Steel (40.6%) showed the highest MAPE values, indicating significant inconsistencies in predictions. The Apple Watch, regardless of body position, was the only activity device with percentage difference CIs crossing zero, indicating no systematic bias toward underestimating or overestimating the true step counts. In contrast, the other 11 devices consistently underestimated the true step counts (Fig 5).

Fig 7 displays a heatmap showcasing the normalised performance scores, including percentage difference, MOE, and MAPE, for each activity device, irrespective of body position. The performance of activity devices exhibited substantial variability across metrics, with pronounced distinctions between the highest- and lowest-performing devices. Based on normalised performance scores, the Apple Watch unequivocally demonstrated superior performance, achieving scores of 1.00 across all metrics (percentage difference, MOE, and MAPE) (Fig 7). This highlights its unmatched accuracy, precision, and reliability. Its performance significantly exceeded that of the second- and third-ranked devices. The Samsung Active demonstrated commendable performance with an average normalised score of 0.80, while the Garmin Vivofit scored 0.79, both achieving reasonable outcomes yet falling considerably short of the Apple Watch. These findings establish the Apple Watch as the most robust and reliable device for tracking walking activity. In contrast, the GENEActiv consistently exhibited the poorest performance, with a normalised score of 0.00 across all metrics, reflecting inherent limitations in its tracking capabilities. Similarly, the Withings Move underperformed, particularly in percentage difference, with a normalised score of 0.06, reflecting poor accuracy. Notably, the ActiGraph wGT3X-BT displayed metric-specific trade-offs, demonstrating moderate precision in MOE (0.72) but showing substantial inaccuracies in percentage difference (0.25), underscoring challenges in accuracy despite reasonable precision.

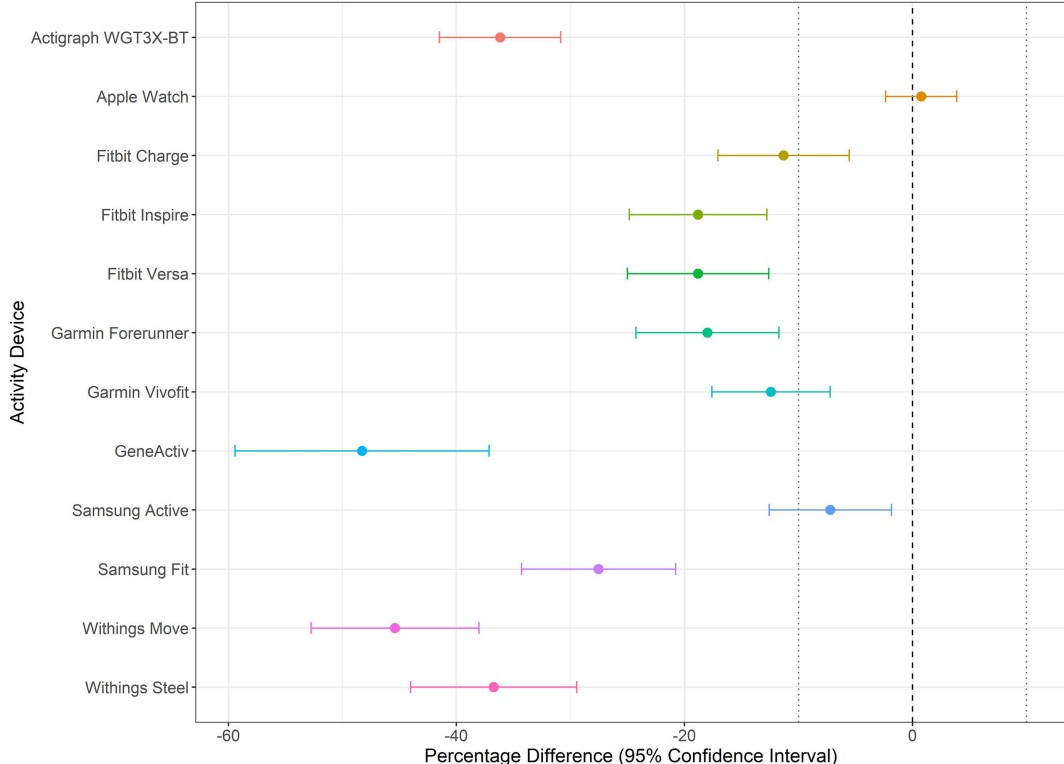

**Fig 5. Percentage difference between device estimated and true step counts, all study conditions.** Vertical dashed lines indicate 10% acceptance threshold.

Pairwise comparisons of estimated marginal means from the linear mixed-effects model revealed that the Apple Watch recorded significantly higher step counts (estimated vs. true steps) than the Garmin Vivofit, Fitbit Charge, and Samsung Active when collapsed across position and walking activity (Fig 8). Mean differences were 46.6 (95% CI: 27.8 to 65.6), 48.2 (95% CI: 26.9 to 67.2), and 28.9 (95% CI: 7.2 to 50.5) steps, respectively. Given the Apple Watch's previously demonstrated alignment with true step counts (i.e., lowest MAPE), its higher estimated counts relative to the other devices reflect better accuracy—not overestimation, or rather a closer match to the manual standard (Fig 8). In contrast, the Garmin, Fitbit, and Samsung devices consistently underestimated step counts. The lack of significant differences among these three (with all pairwise confidence intervals crossing zero) suggests they performed similarly and less accurately than the Apple Watch. Collectively, these results indicate that the Apple Watch provides a significantly more accurate estimate of step count, while the other devices tend to underestimate steps and perform comparably to each other.

## Discussion

The findings of this study demonstrated that the Apple Watch outperformed all other devices across performance metrics. While it performed best when positioned on the waist, it also exhibited good accuracy and adaptability (interchangeability) when placed on the arm or leg, highlighting its superior suitability for application on hospital patients, where limb access can be compromised by dressings, casts or vascular devices and this may change. Additionally, the Fitbit Charge, Fitbit Inspire, and Fitbit Versa performed strongly when positioned on the leg, as did the Garmin Vivofit when positioned on the arm. Interestingly, these devices demonstrated less adaptability to changes in position compared to the Apple Watch, and performance significantly declined when worn in alternative locations. These findings underscore the importance of

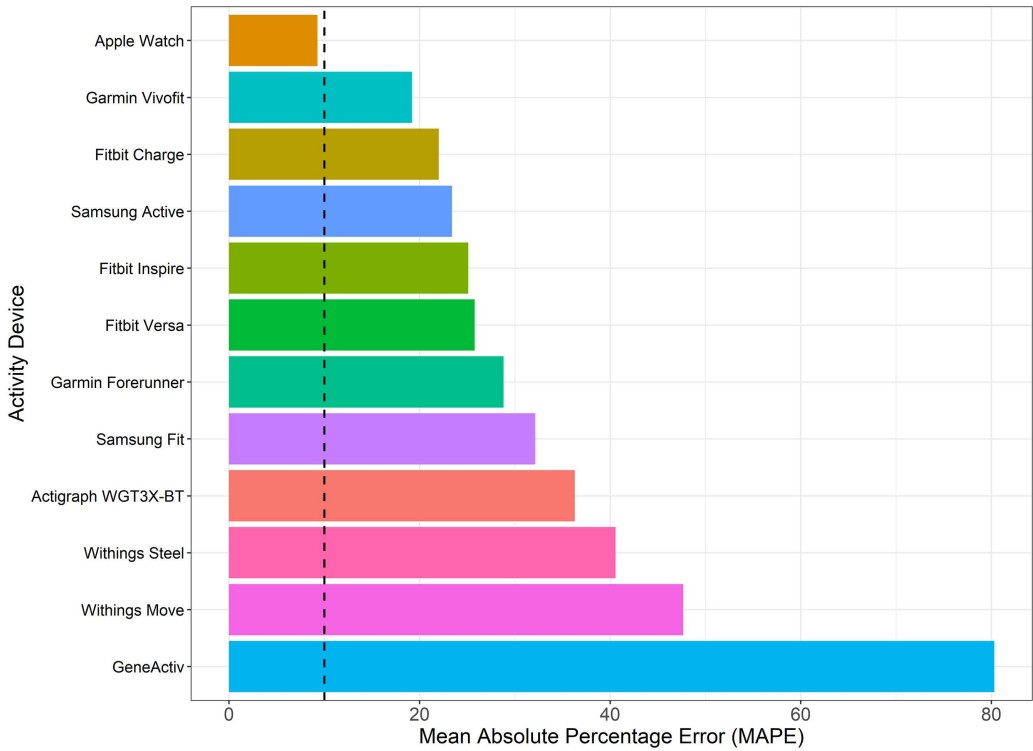

**Fig 6. Mean absolute percentage error (MAPE) between device estimated and true step counts, all study conditions.** Vertical dashed lines indicate 10% acceptance threshold.

considering both the sensor technology being deployed by each device and its adaptability and placement when seeking to measure step count as a marker of physical activity, using monitoring devices. The study findings also highlight that planning is crucial to measure physical activity accurately in populations where movement; gait speed, and, or limb access are compromised, particularly as wear location may need to vary with each application or repeat session. Lastly, several devices performed poorly regardless of their placement on the body, reflecting inherent limitations in their tracking capabilities during slow or altered walking and, or post-hoc processing methods.

## Activity device performance

The Apple Watch was the most accurate and reliable activity device for tracking slow and altered walking, particularly when positioned on the waist. Its superior performance may be attributed to advanced sensor technology and sophisticated algorithms, consistent with prior research demonstrating its reliability in step counting. For example, Bai et al. (2021) assessed the accuracy of the Apple Watch (worn on the arm) in estimating daily step counts in a free-living setting and found it to be comparably accurate to their criterion device, the Yamax Digi-Walker SW-200 (worn on the waist) [37]. Another notable finding was the consistent performance of the Apple Watch across all body positions (arm, waist, leg), with a MAPE score below 10% when data were collapsed across positions, indicating high predictive accuracy and interchangeability. This makes it a highly versatile option for tracking motion in clinical patients with impaired walking and limited options for device placement (Fig 6). Interestingly, the Fitbit Charge, Fitbit Inspire, and Fitbit Versa performed best when positioned on the leg, rather than on the arm, despite Fitbit's claims of superior step-count tracking when worn on the arm as recommended (Fig 4) [38]. A systematic review similarly found that Fitbit devices provide accurate step counts

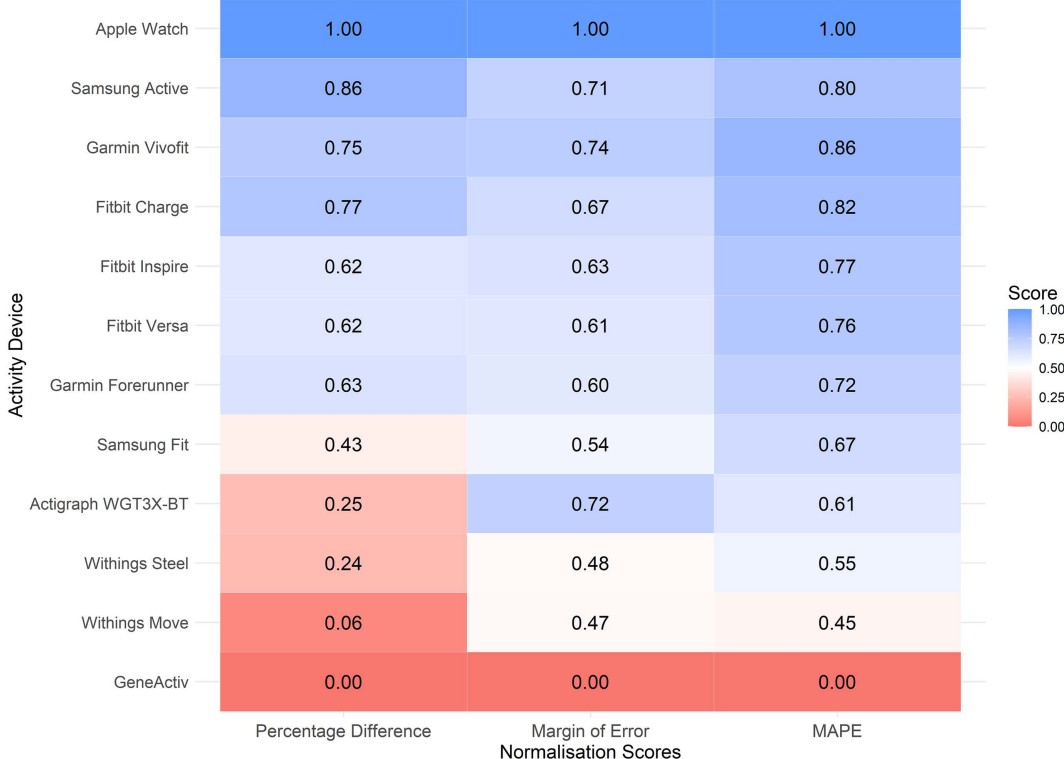

**Fig 7. Device performance metrics heatmap, all study conditions combined.** Normalised scores (0-1) for: percentage difference (left), margin of error [MOE](middle), and mean absolute percentage error [MAPE](right). Colour shading: poorer (red); moderate (white); and, higher (blue) normalised performance scores. Device rank reflects comparative overall performance.

during slow or very slow walking when placed on the ankle in adults without mobility impairments [39]. Collectively, these findings suggest that Fitbit devices offer moderate to high accuracy and reliability for step tracking when worn on the leg (ankle), particularly during slow or altered walking, highlighting their potential utility in hospital settings.

The GENEActiv device's poor performance in detecting slow or altered walking raises concerns about its suitability for hospital settings. Similarly, Mora-Gonzalez et al. [40] found that at speeds of 0.8–1.6 km/h, the GENEActiv exhibited MAPE values of approximately 20–30%, highlighting difficulties in step detection during low-intensity or low speed activities [40,41]. While these limitations may stem from the device's sensor sensitivity and calibration—where oversampling during slow or altered movement could amplify noise and obscure true step events, further reducing accuracy—its poor performance in the present study likely reflects multiple factors, including also the data processing method used to determine step count. These findings underscore broader challenges clinicians and researchers face when using step-based metrics. Many step detection algorithms require predefined, often poorly defined, parameters, such as acceleration thresholds and activity mode, before processing data [42]. In the case of the GENEActiv device, the absence of native onboard step detection necessitates external data processing. In our study, as noted earlier,.bin files were analysed using the 'GENEAclassify' package in R, applying the 'getGENEAsegments()' function. This approach was intentionally selected as a simple, transparent, and reproducible method, designed to facilitate future uptake by clinicians and researchers who may not have advanced programming skills. Notably, a similar approach was employed by Schoenfelder et al. [43], who processed GENEActiv.bin files using the 'GENEAclassify' package (v1.5.1) to derive stepping metrics from raw accelerometer data. While their pipeline incorporated variable-length event segmentation, the underlying analytical

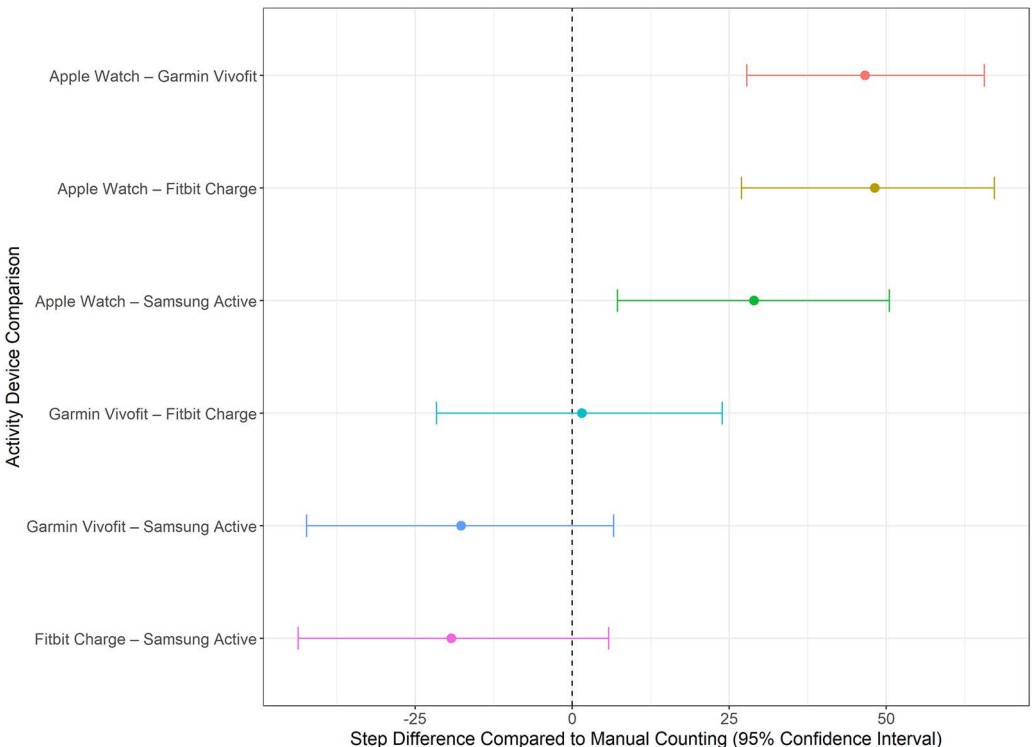

**Fig 8. Bootstrapped pairwise comparisons of step count differences between activity devices.** Analyses assess overall device performance irrespective of body position, accounting for within-subject dependency. Confidence intervals that do not cross zero indicate significant differences.

framework—open-source, R-based, and reliant on post hoc signal processing—aligns closely with the method adopted in our study. Their validation of the 'GENEAcount' algorithm, which showed excellent agreement with activPAL° for total daily steps (concordance correlation coefficient = 0.88), supports the potential for low-complexity approaches to yield reliable step count data when appropriately implemented. While our method intentionally reduced the complexity of data interaction, this likely limited the algorithmic sophistication of step detection and contributed to the observed inaccuracies in the walking conditions. We acknowledge that more advanced signal processing techniques—such as machine learning classifiers—have shown high reliability for step detection using GENEActiv data [44]. Therefore, variability in step count accuracy across studies reflects not only device characteristics but also the diversity in analytical pipelines. These findings underscore the need for greater standardisation and validation of processing methods and highlight the trade-off between end user accessibility and analytical precision in clinical research workflows [45].

The step count performance metrics of device-position combinations varied significantly, highlighting trade-offs that must be considered for specific applications. For example, the Garmin Forerunner, when positioned on the leg, performed well in terms of percentage difference, indicating low deviation from the reference standard with reasonable accuracy. However, it exhibited poorer MOE and MAPE scores, reflecting reduced consistency between sessions and predictive reliability. This suggests that while the device-position combination can provide relatively accurate step estimates on average, its variability and errors in individual predictions may limit its suitability for applications where changes of body position may be necessary by session or high consistency and reliability. These findings underscore the importance of aligning device selection and positioning with the specific demands of the task [39,40,46]. Understanding these trade-offs allows for more informed decisions, ensuring that devices and their placements are tailored to meet the diverse needs of users and applications.

Only three of the top 20 device-position combinations for overall normalised scores, which account for both accuracy and precision, involved the waist. This underscores the high variability in measurements obtained from this location, despite the Apple Watch at the waist being the best-performing combination. This variability suggests that the reliability of the waist as a placement for activity monitoring devices may be less reliable across different devices and algorithms. It also must be acknowledged that most of the devices tested are not specifically designed for mounting at the waist. This finding contrasts with much of the existing literature, which frequently identifies the waist as a reliable site for step counting due to its proximity to the body's centre of mass and its capacity to capture whole-body motion, making it well-suited for digital wearables [47–49]. The variability observed in this study may also reflect unknown device-specific algorithmic limitations or sensitivity to individual movement patterns, such as torso rotation or irregular gaits. These results challenge the assumption that the waist is an accurate and reliable position for tracking activity, especially in populations with compromised mobility. Notably, the leg accounted for 10 of the top 20 device-position combinations in overall normalised scores, suggesting larger amplitude arcs of movements may offer a more consistent and reliable alternative for step counting across diverse applications. In fact, midsole-worn pedometers have demonstrated superior accuracy and reliability in step counting across various activities, including walking, compared to wrist-worn devices [47]. These findings underscore the need for further research to investigate the potential benefits of leg placement for activity monitoring, particularly in clinical populations with compromised walking patterns.

The Apple Watch showed no systematic bias across all body positions, with confidence intervals (CIs) for its step count estimates extending beyond zero when data was collapsed across positions. This indicates that the device neither consistently overestimated nor underestimated step counts, underscoring its reliability and adaptability across varied wear locations. In contrast, as was expected with the imposed slowed body movements, all other devices systematically underestimated true step counts, with CIs consistently below zero. This pattern held across pairwise comparisons, with no significant differences in underestimation among the Garmin Vivofit, Fitbit Charge, and Samsung Active — all of which were significantly less accurate than the Apple Watch. These four devices were selected based on having the lowest mean absolute percentage error (MAPE) scores. Collectively, these findings highlight a shared limitation among the non-Apple devices in detecting steps under slow or irregular walking conditions. This trend aligns with findings from previous studies, which have noted a tendency for some devices to underestimate steps during controlled testing, such as treadmill walking [39] and slow walking speeds [17]. This underestimation poses significant challenges in clinical or hospital applications, where accurate activity monitoring is essential for assessing unobserved mobility and, or tracking progress of the volume of physical activity. When considering the impact of body position, 26 out of the 36 device-position combinations, including all 12 involving the arm, systematically underestimated step counts. In contrast, 5 of the 6 combinations that showed no systematic bias were positioned on the leg, suggesting that the leg may provide a more reliable placement for accurate step monitoring in clinical applications. To overcome the observed limitations, calibration and algorithm refinement are essential, particularly to accommodate diverse use cases, such as individuals with compromised mobility or irregular gait patterns. These advancements would improve device reliability and applicability in real-world scenarios, enabling accurate and meaningful activity monitoring across diverse populations and settings.

## Practical implications

In addition to considering device costs, evaluating the practical implications of accuracy and precision is critical when selecting activity devices for specific clinical or research applications. Devices like the Apple Watch, which excel in both metrics, are particularly well-suited for scenarios requiring detailed activity monitoring, such as rehabilitation or high-stakes healthcare settings. This underscores the broader potential of consumer-grade devices in clinical and research contexts, where accessible and adaptable tools are essential. However, no single device is universally optimal; performance depends on the specific metric, placement, and user population characteristics. For clinicians, these findings offer valuable guidance in selecting devices tailored to patients' needs, particularly those with mobility

impairments or atypical movement patterns. Manufacturers should focus on optimising algorithms for specific body positions, as performance varies significantly by wear location [50]. This is especially important for clinical patients, where irregular movements or limited access to common placements, such as the wrist, may affect device usability. Additionally, integrating adaptive algorithms to account for variations in movement patterns, including slower cadences or asymmetrical gaits often observed in clinical populations, could enhance device accuracy [51]. Designing devices with versatile attachment options and compatibility with mobility aids could further improve usability and precision [52]. Addressing these considerations would enable more reliable and meaningful activity monitoring across diverse contexts.

## Strengths and limitations

A strength of this study lies in the comprehensive evaluation of devices across multiple metrics and body positions, offering a nuanced understanding of their performance under standardised, though simulated, hospital patient conditions. This approach allows for a detailed comparison of device accuracy, precision, and robustness, ensuring that key performance aspects are systematically assessed. However, the study's focus on simulated, controlled conditions may limit generalisability of findings to real-world applications, where variability in walking patterns, environmental factors, and patient-specific conditions is more pronounced, particularly in complex hospital settings. During preliminary testing using early generation movement sensors in a burn patient population, it was observed that reducing gait speed was associated strongly with increasing missing step count data (unpublished activPAL® data, Edgar DW, Chen A, Wood FW, 2012). Thus, to reduce confounders in this study, testing was conducted on self-reported healthy individuals who simulated altered, and specifically slow, walking patterns. It is acknowledged, that while useful for preliminary evaluations, our 'structured, reproducible' movement adjustments may not fully replicate the biomechanical complexities of patients with clinical gait abnormalities. In addition, due to the significant time burden on participants to complete the study tasks, we did not include control testing with faster and, or ostensibly 'normal' walk speeds. While evidence exists evaluating device performance in clinical populations [53,54], future research should build on this by further exploring their performance in specific groups, such as patients with trauma-induced gait impairments, neurological conditions, or post-surgical recovery [55]. An additional limitation of this study is the lack of objective constraint on the shuffling gait condition. Participants were given standardised verbal instructions and a physical demonstration to guide performance. However, the study, conducted in clinical area, did not have access to motion capture or inertial sensors which could have provided data to monitor or enforce specific joint or movement parameters. This, with the diverse anthropometry of the participants, may have introduced variability in gait characteristics, potentially affecting step detection accuracy. However, there was an intentional trade-off to enhance the ecological validity of the study, as our aim was to evaluate device performance under conditions that reflect real-world clinical environments. Future studies may benefit from incorporating objective measures or biofeedback to improve reproducibility of research conditions, while maintaining relevance to clinical populations. It is further acknowledged that participant's natural walk speed and body habitus measures were not captured during the study. These details may have been useful as an adjustment in statistical modelling to reduce interparticipant variations. However, the adding of another walk condition and taking personal measurements, none of which were directly aligned with the primary aim, was considered unacceptable additional burdens for participants that may impact recruitment. Those who committed to the study devoted significant time and therefore, these extras were not included in the protocol. These future study adjustments would provide more comprehensive insights into the devices' accuracy, reliability, and practical applications in real-world healthcare scenarios, particularly for monitoring and supporting patient rehabilitation. Lastly, the lack of blinding of participants to the device brands is acknowledged, however, it was considered too challenging to implement when weighed against the strategies in place to mitigate any personal-based bias introduction, which included randomisation of gait condition, body location and 12 devices applied judiciously for each participant.

## Conclusion

This study demonstrated the marked variability in activity device performance in measuring step counts during altered, slowed ambulation. The Apple Watch demonstrated superior performance across consumer-defined, pre-determined metrics, showcasing its reliability, and adaptability across conditions and interchangeability between body locations. The study contributed new knowledge to inform design of programs and research to monitor physical activity in populations and scenarios where movement, particularly ambulation, is compromised, such as in hospital or after injury.

## Supporting information

**S1 File. Details of devices.**
(DOCX)

## Author contributions

**Conceptualization:** David Weight, Alethea Rea, Jenny Conlon, Fiona M. Wood, Dale W. Edgar.

**Data curation:** David Weight.

**Formal analysis:** Grant Rowe, David Weight.

**Funding acquisition:** David Weight, Fiona M. Wood.

**Investigation:** David Weight.

**Methodology:** David Weight, Alethea Rea, Jenny Conlon, Fiona M. Wood, Dale W. Edgar.

**Project administration:** David Weight, Dale W. Edgar.

**Resources:** Fiona M. Wood, Dale W. Edgar.

**Software:** David Weight.

**Supervision:** Alethea Rea, Jenny Conlon, Fiona M. Wood, Dale W. Edgar.

**Visualization:** Grant Rowe.

**Writing – original draft:** Grant Rowe, Jenny Conlon, Fiona M. Wood, Dale W. Edgar.

**Writing – review & editing:** Grant Rowe, David Weight, Alethea Rea, Jenny Conlon, Fiona M. Wood, Dale W. Edgar.

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
