## [Decision Letter · Decision Letter 0]

8 Jul 2025

Dear Dr. Edgar,

Thank you for submitting your manuscript to PLOS ONE. After careful consideration, we feel that it has merit but does not fully meet PLOS ONE’s publication criteria as it currently stands. Therefore, we invite you to submit a revised version of the manuscript that addresses the points raised during the review process.

We look forward to receiving your revised manuscript.

Kind regards,

Yih-Kuen Jan, PhD

Academic Editor

PLOS ONE

Journal Requirements:

3. In the online submission form, you indicated that [Data cannot be shared publicly because of Ethics approval provisions. Data are available from the SMHS Institutional Data Access / Ethics Committee (contact via SMHS.rgo@health.wa.gov.au) for researchers who meet the criteria for access to confidential data.].

5. Please include captions for your Supporting Information files at the end of your manuscript, and update any in-text citations to match accordingly. Please see our Supporting Information guidelines for more information: http://journals.plos.org/plosone/s/supporting-information .

Reviewers' comments:

Reviewer's Responses to Questions

**Comments to the Author**

1. Is the manuscript technically sound, and do the data support the conclusions?

Reviewer #1: Partly

Reviewer #2: Yes

2. Has the statistical analysis been performed appropriately and rigorously?

Reviewer #1: No

Reviewer #2: No

3. Have the authors made all data underlying the findings in their manuscript fully available?

Reviewer #1: Yes

Reviewer #2: Yes

4. Is the manuscript presented in an intelligible fashion and written in standard English?

Reviewer #1: Yes

Reviewer #2: Yes

Reviewer #1: This study evaluates the accuracy of twelve wearable devices in detecting step counts during slow and shuffling walking, simulating hospital conditions. The Apple Watch, particularly when worn on the waist, demonstrated the highest accuracy and consistency across all metrics. The experimental design was rigorous, with standardized gait conditions, manual video-based step counting, and comprehensive testing across arm, waist, and leg placements. The use of healthy participants simulating altered gait is a practical limitation, as it may not fully replicate the complexity of real patient movement. In addition, while the Apple Watch's strengths are emphasized, the reasons behind poor performance of other devices, such as GENEActiv, are less thoroughly examined. The focus on only slow and shuffling gaits also limits the generalizability to broader walking patterns. Nonetheless, the findings have strong clinical relevance. They support using the Apple Watch for activity monitoring in settings with impaired gait, and highlight the importance of body placement for accurate measurement. These insights can inform clinical practice, wearable device design, and algorithm development for more robust patient monitoring.

A few minor issues need to be addressed before publication.

1. Lack of real patient data: Although using healthy participants to simulate slow and shuffling gait is a reasonable starting point, the absence of clinical populations (e.g., elderly or neurologically impaired patients) limits the external validity. Simulated gait may not capture biomechanical and neuromuscular irregularities, such as asymmetrical patterns or compensatory strategies, that influence wearable device performance in actual hospital settings.

2. Underexplored device-specific limitations: The poor performance of devices like GENEActiv and Withings is reported, but not adequately analyzed. A more technical discussion is needed regarding the potential causes—such as accelerometer placement sensitivity, sampling frequency, lack of native step detection algorithms, or preprocessing methods—to provide useful guidance for both researchers and developers.

3. Unclear randomization protocol for device-body placement: The manuscript mentions rotating devices across the arm, waist, and leg, but does not clarify whether a consistent or randomized sequence was used for device positioning across participants. If the order was fixed or unbalanced, it may introduce systematic bias due to fatigue or acclimation effects. Details about the randomization or counterbalancing strategy are necessary to validate the internal control of the experimental design.

4. Insufficient constraint on shuffling gait standardization: The “shuffle walk” condition is described as reduced hip and knee flexion, but no objective control or instrumentation (e.g., motion capture thresholds, wearable inertial measures) was reported to enforce consistency. Since shuffling can vary widely across individuals, lack of constraint may introduce significant variability in motion amplitude and step detection, affecting device comparability and statistical power. More standardized control (e.g., minimum flexion angles or visual feedback) is needed to ensure reproducibility.

5. Statistical modeling of repeated measures: The study includes repeated walking blocks and multiple device placements per participant, yet the manuscript does not specify whether statistical models (e.g., linear mixed models) were used to account for within-subject dependency. Failing to model repeated measures appropriately could inflate type I errors or underestimate variance, affecting interpretation of device performance rankings.

Reviewer #2: Dear Authors,

Your manuscript proposes a hybrid deep learning model combining Stacked LSTM and Random Forest for the early detection of liver cirrhosis using clinical data. This is an important area of research, and your focus on using machine learning for early diagnosis is timely and relevant.

To improve the quality and impact of your work, I offer the following suggestions:

(a) Clarify the rationale for using SLSTM: Please explain why a recurrent architecture is appropriate for this dataset, which appears to be primarily tabular/non-temporal.

(b) Benchmark with additional models: Include comparisons with other machine learning techniques (e.g., XGBoost, Logistic Regression, CNN for tabular data) and/or clinical scores (e.g., MELD or Child-Pugh scores if relevant).

(c) Add statistical rigor: Incorporate cross-validation, report standard deviations, and if possible, include statistical significance testing to support the claims.

(d) Improve interpretability: Include SHAP or similar methods to show which features contribute most to the model’s predictions. Feature importance is shown for Random Forest, but interpretability is not explored (e.g., SHAP, LIME), which is essential for healthcare applications.

(e) Language and formatting: The manuscript would benefit from language polishing and clearer figure/table labeling. For example, figure captions should be more descriptive.

(f) Share processing pipeline: Consider providing your detailed preprocessing steps to enhance transparency and reproducibility.

**Do you want your identity to be public for this peer review?** For information about this choice, including consent withdrawal, please see our Privacy Policy

Reviewer #1: No

Reviewer #2: **Yes: ** Shake Ibna Abir

---

## [Decision Letter · Decision Letter 1]

31 Aug 2025

Dear Dr. Edgar,

Thank you for submitting your manuscript to PLOS ONE. After careful consideration, we feel that it has merit but does not fully meet PLOS ONE’s publication criteria as it currently stands. Therefore, we invite you to submit a revised version of the manuscript that addresses the points raised during the review process.

We look forward to receiving your revised manuscript.

Kind regards,

Yih-Kuen Jan, PhD

Academic Editor

PLOS ONE

Journal Requirements:

Reviewers' comments:

Reviewer's Responses to Questions

**Comments to the Author**

Reviewer #1: All comments have been addressed

Reviewer #3: All comments have been addressed

2. Is the manuscript technically sound, and do the data support the conclusions?

Reviewer #1: Yes

Reviewer #3: Partly

3. Has the statistical analysis been performed appropriately and rigorously?

Reviewer #1: Yes

Reviewer #3: I Don't Know

4. Have the authors made all data underlying the findings in their manuscript fully available?

Reviewer #1: Yes

Reviewer #3: Yes

5. Is the manuscript presented in an intelligible fashion and written in standard English?

Reviewer #1: Yes

Reviewer #3: Yes

Reviewer #1: The authors have provided a thorough and thoughtful revision of their manuscript, and I thank them for their diligent work in addressing the concerns raised in the initial review. The authors have corrected the major methodological issue concerning the statistical analysis by implementing a linear mixed-effects model to appropriately account for repeated measures. The addition of this robust analysis, including pairwise comparisons and a new Figure 8, has significantly strengthened the validity of the study's conclusions. The manuscript is now a much stronger and more rigorous piece of research. The authors have successfully addressed all my previous concerns.

Reviewer #3: In this study, the authors have investigated the accurate measurement of simulated slow and altered walking activity: Apple Watch best in class wearable devices. The topic of the manuscript is interesting. However, there are some minor concerns that should be addressed before the article can be accepted for publication.

1. Please add more subjects’ information, ex. age, BMI etc.

2. Explain the experimental procedure, for example. Is the order of the walking pattern random? Is the order of wearable devices also random ? Do these subjects know which device brand of device they wear while experiments?

3. Have the authors considered letting these subjects walk at their own speed?

4. Please explain more specifications about the device, for example, manufacture date, tolerance range, and even price. If the devices need to be calibrated, are the settings correct?

5. The resolution of the figures needs to be improved.

6. Part of the title “Apple Watch best in class wearable devices.” It might not be suitable for an academic research article. If it is possible, maybe it needs to be modified.

**Do you want your identity to be public for this peer review?** For information about this choice, including consent withdrawal, please see our Privacy Policy

Reviewer #1: **Yes: ** Pu-Chun Mo

Reviewer #3: No

---

## [Author Response · Author response to Decision Letter 2]

11 Sep 2025

Thank you to the reviewers for taking the time to review our revised manuscript - please see extended responses in the attached document.

---

## [Decision Letter · Decision Letter 2]

15 Sep 2025

Accurate measurement of simulated slow and altered walking activity: Apple Watch best in class wearable devices

PONE-D-25-18604R2

Dear Dr. Edgar,

We’re pleased to inform you that your manuscript has been judged scientifically suitable for publication and will be formally accepted for publication once it meets all outstanding technical requirements.

Kind regards,

Yih-Kuen Jan, PhD

Academic Editor

PLOS ONE

Additional Editor Comments (optional):

Reviewers' comments:

Reviewer's Responses to Questions

**Comments to the Author**

Reviewer #3: All comments have been addressed

2. Is the manuscript technically sound, and do the data support the conclusions?

Reviewer #3: Yes

3. Has the statistical analysis been performed appropriately and rigorously?

Reviewer #3: Yes

4. Have the authors made all data underlying the findings in their manuscript fully available?

Reviewer #3: Yes

5. Is the manuscript presented in an intelligible fashion and written in standard English?

Reviewer #3: Yes

Reviewer #3: (No Response)

**Do you want your identity to be public for this peer review?** For information about this choice, including consent withdrawal, please see our Privacy Policy

Reviewer #3: No

---

## [Editor Report · Acceptance letter]

PONE-D-25-18604R2

PLOS ONE

Dear Dr. Edgar,

I'm pleased to inform you that your manuscript has been deemed suitable for publication in PLOS ONE. Congratulations! Your manuscript is now being handed over to our production team.

Kind regards,

on behalf of

Dr. Yih-Kuen Jan

Academic Editor

PLOS ONE